# Crossover in aromatic amino acid interaction strength between tyrosine and phenylalanine in biomolecular condensates

**David De Sancho\*, Xabier Lopez\***

Polimero eta Material Aurreratuak: Fisika, Kimika eta Teknologia, Kimika Fakultatea, UPV/EHU & Donostia International Physics Center (DIPC), Donostia / San Sebastian, Spain

## eLife Assessment

This **important** study uses advanced computational methods to elucidate how environmental dielectric properties influence the interaction strengths of tyrosine and phenylalanine in biomolecular condensates. The evidence supporting the claims of the authors is **convincing**, as the simulations are performed rigorously providing mechanistic insights into the origin of the differences between the two aromatic amino acids considered. This study will be of broad interest to researchers studying biomolecular phase separation.

**\*For correspondence:**
david.desancho@ehu.eus (DDS);
xabier.lopez@ehu.eus (XL)

**Competing interest:** The authors declare that no competing interests exist.

## Abstract

Biomolecular condensates often form through the self-assembly of disordered proteins with low-complexity sequences. In these polypeptides, the aromatic amino acids phenylalanine and tyrosine act as key 'sticker' residues, driving the cohesion of dense phases. Recent studies on condensates suggest a hierarchy in sticker strength, with tyrosine being more adhesive than phenylalanine. This hierarchy aligns with experimental data on amino acids solubilities and potentials of mean force derived from atomistic simulations. However, it contradicts conventional chemical intuition based on hydrophobicity scales and pairwise contact statistics from experimental structures of proteins, which suggest that phenylalanine should be the stronger sticker. In this work, we use molecular dynamics simulations and quantum chemistry calculations to resolve this apparent discrepancy. Using simple model peptides and side-chain analogues, we demonstrate that the experimentally observed hierarchy arises from the lower free energy of transfer of tyrosine into the condensate, mediated by both stronger protein-protein interactions and solvation effects in the condensate environment. Notably, as the dielectric constant of the media surrounding the stickers approaches that of an apolar solvent, the trend reverses, and phenylalanine becomes the stronger sticker. These findings highlight the role of the chemical environment in modulating protein-protein interactions, providing a clear explanation for the crossover in sticker strength between tyrosine and phenylalanine in different media.

## Introduction

For decades, cellular organization was primarily understood through the lens of membrane-bound compartmentalization. However, it has become increasingly clear that membrane-less organelles — such as stress granules, nuclear speckles, and Cajal bodies— are also widespread within cells (*Banani et al., 2017*). At least in some cases, these organelles form by phase separation of their components

(*Shin and Brangwynne, 2017*), primarily disordered regions of proteins and nucleic acids, which partition into dense and dilute phases. The condensates thus formed retain properties of liquids, like fusion, dripping, and wetting (*Brangwynne et al., 2009*). This phenomenon has generated significant interest in the physical mechanisms governing the phase behaviour of biomolecular mixtures (*Choi et al., 2020*).

Many proteins undergoing phase separation share common characteristics, including deviations from typical compositions of folding proteins and simple sequences with multiple amino acids repeats. Specifically, in the sequences of proteins that experience upper critical solution temperature (UCST) transitions, like the low-complexity regions of FUS, hnRNPA1, Ddx4, LAF-1, or atGRP7, we observe long stretches rich in polar residues interspersed with aromatics or positively charged residues (*Martin and Mittag, 2018*). Borrowing language from the theory of associative polymers (*Semenov and Rubinstein, 1998*), these units of sequence have been termed 'stickers,' in the case of aromatics and positively charged residues, and 'spacers,' for the polar residue repeats (*Martin et al., 2020*; *Bremer et al., 2022*; *Mittag and Pappu, 2022*). Spacers act as linkers that lend flexibility to the polypeptide mesh in the protein-dense phase. In contrast, stickers play a key role in determining both the single-chain properties of the polymer and the phase behaviour of the condensate through interactions involving their aromatic or charged groups (*Wang et al., 2018*). Recent experiments have quantified the influence of different types of stickers on the polymer properties and phase behaviour (*Bremer et al., 2022*). This raises a crucial question: what is the fundamental origin of the relative strengths of different stickers?

This matter has recently been investigated in the context of the cationic amino acids lysine (Lys) and arginine (Arg) (*Wang et al., 2018*; *Das et al., 2020*; *Schuster et al., 2020*; *Fisher and Elbaum-Garfinkle, 2020*; *Greig et al., 2020*; *Paloni et al., 2021*; *Hong et al., 2022*). Despite having the same net charge, these residues turn out not to be interchangeable. Mutagenesis experiments on LAF-1 have shown that substituting Arg with Lys completely suppresses phase separation (*Schuster et al., 2020*). This distinction is functionally relevant, as Arg to Lys substitutions affect speckle formation (*Greig et al., 2020*). Additionally, experiments on the intrinsically disordered region (IDR) of Ddx4 indicate that phase separation is favoured by Arg relative to Lys (*Brady et al., 2017*; *Das et al., 2020*; *Schuster et al., 2020*). Das and co-workers attempted to explain arginine's greater propensity to phase separate in Ddx4 variants using coarse-grained simulations with two different energy functions (*Das et al., 2020*). The model was first parametrized using a hydrophobicity scale, aimed to capture the 'stickiness' of different amino acids (*Dignon et al., 2018*), but this did not recapitulate the correct rank order in the stability of the simulated condensates (*Das et al., 2020*). By replacing the hydrophobicity scale with interaction energies from amino acid contact matrices —derived from a statistical analysis of the PDB (*Dignon et al., 2018*; *Miyazawa and Jernigan, 1996*; *Kim and Hummer, 2008*)— they recovered the correct trends (*Das et al., 2020*). A key to the greater propensity to phase separate in the case of Arg may derive from the pseudo-aromaticity of this residue, which results in a greater stabilization relative to the more purely cationic character of Lys (*Gobbi and Frenking, 1993*; *Wang et al., 2018*; *Hong et al., 2022*).

Here, we focus on the distinct roles of the main aromatic residues acting as stickers —tyrosine (Tyr) and phenylalanine (Phe)— excluding tryptophan due to its much lower abundance (*Maraldo et al., 2024*). Given that they only differ in a hydroxyl group, one could expect Tyr and Phe to be equally relevant to phase separation, especially considering the finding from a double-mutant cycle that Tyr-Tyr and Phe-Phe pairs make nearly identical contributions to protein stability (*Serrano et al., 1991*). However, experimental evidence on various proteins suggests that Tyr is a stronger driver of condensation (*Lin et al., 2017*; *Wang et al., 2018*; *Schuster et al., 2020*; *Bremer et al., 2022*). This is demonstrated by the reduced propensity to phase separate of Tyr-to-Phe mutants of FUS and LAF-1 (*Lin et al., 2017*; *Wang et al., 2018*; *Schuster et al., 2020*) and the opposite effect in Phe-to-Tyr mutants of hnRNPA1 (*Wang et al., 2018*; *Bremer et al., 2022*). Understanding the origin of the different sticker strengths of Phe and Tyr are important due to their functional relevance. In a large sample of hnRNPA1 variants, the fractions of Phe and Tyr have been found to co-vary, suggesting that evolutionary control of composition fine-tunes the properties of condensates (*Bremer et al., 2022*).

The molecular properties of Phe and Tyr may give important insights about their distinct behaviour as stickers in condensates. Hydrophobicity scales typically rank Phe as the most hydrophobic residue (*Kyte and Doolittle, 1982*; *Tesei et al., 2021*), consistent with the greater hydration free energy

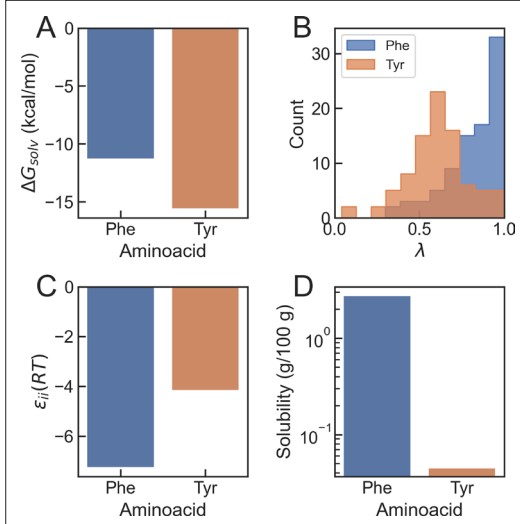

**Figure 1.** Bibliographic values for different properties of phenylalanine and tyrosine. (**A**) Solvation free energy ($\Delta G_{\text{solv}}$) (*Chang et al., 2007*). (**B**) Probability distributions of min-maxed normalized hydropathy values $\lambda$ from bibliographic hydrophobicity scales (*Tesei et al., 2021*). (**C**) Self-interaction energy ($\varepsilon_{ii}$) from the Miyazawa-Jernigan contact matrix (*Miyazawa and Jernigan, 1996*). (**D**) Solubility in water at 25°C (*Nozaki and Tanford, 1971*).

of Tyr relative to Phe (*Wolfenden et al., 1981*; *Chang et al., 2007*) (see *Figure 1A–B*). As in the case of Arg and Lys, using hydrophobicity as a proxy for interaction energy in bead simulation models proved insufficient to explain the relative strengths of Phe and Tyr as stickers (*Das et al., 2020*). However, in this case, statistical contact matrices also could not capture the correct order of stickiness. In the Miyazawa and Jernigan statistical potential, Tyr-Tyr contacts are weaker than Phe-Phe (with energies of –4.17 and –7.26 in $RT$ units, respectively; see *Figure 1C*; *Miyazawa and Jernigan, 1996*). We note that a different hydrophobicity scale based on peptides undergoing inverse temperature transitions —i.e., the Urry scale (*Urry et al., 1992*), where Tyr is more hydrophobic— can account for the correct rank order in saturation concentration (*Regy et al., 2021*). On the other hand, the potentials of mean force between amino acids calculated with atomistic force fields have a deeper free energy well for the Tyr-Tyr pair than for Phe-Phe (*Chelli et al., 2002*; *Joseph et al., 2021*). A final data point of interest is the extremely low solubility of Tyr, over an order of magnitude smaller than that of Phe (0.045 and 2.79 g/100 g of water, respectively; see *Figure 1D*; *Nozaki and Tanford, 1971*). This low solubility has led to the suggestion that Tyr hydrogen bonds are stronger in the protein interior than those formed in water (*Pace et al., 2001*).

In summary, experimental results on condensates containing Phe/Tyr variants do not seem to align with either solvation free energies, most hydrophobicity scales, or statistical contact potentials, but are consistent with calculations with atomistic force fields in solution and solubilities in water. One possible explanation for these conflicting findings is that, due to their level of hydration, molecular condensates may differ significantly from the tightly-packed cores of folded protein structures (*Lin et al., 2017*; *Das et al., 2020*). Here, we address this paradox using a combination of classical molecular dynamics (MD) simulations and quantum chemical calculations. First, we estimate transfer free energies of peptides including these aromatic residues into model peptide condensates, and find that they are more favourable for Tyr than for Phe, an effect that is reversed when we perform the same transformation in apolar media. DFT calculations confirm that the interaction energies in Tyr-Tyr pairs are stronger than those between Phe residues. However, the transfer free energy contribution dominates at sufficiently low dielectric constants, making Phe-Phe pairs more favourable. These findings recapitulate the right rank order of interaction strengths of aromatic stickers in biomolecular condensates, but also their crossover in low dielectric media like the hydrophobic cores of folded proteins.

## Methods
### Classical molecular dynamics simulations
#### Molecular models
In this work, we report simulations of terminally-capped GGXGG peptides with X=F/Y in different media, including water, organic solvents, and peptide condensates. This family of peptides has been characterised extensively in the past, both from experiment (*Plaxco et al., 1997*) and simulation (*Workman and Pettitt, 2021*). We have built structures for the peptides using AmberTools (*Salomon-Ferrer et al., 2013*). Our condensates are formed by an equimolar mixture of Gly, Ser and either Tyr or Phe 'dipeptides' (in fact, terminally capped amino acids), as before (*De Sancho, 2022*). For

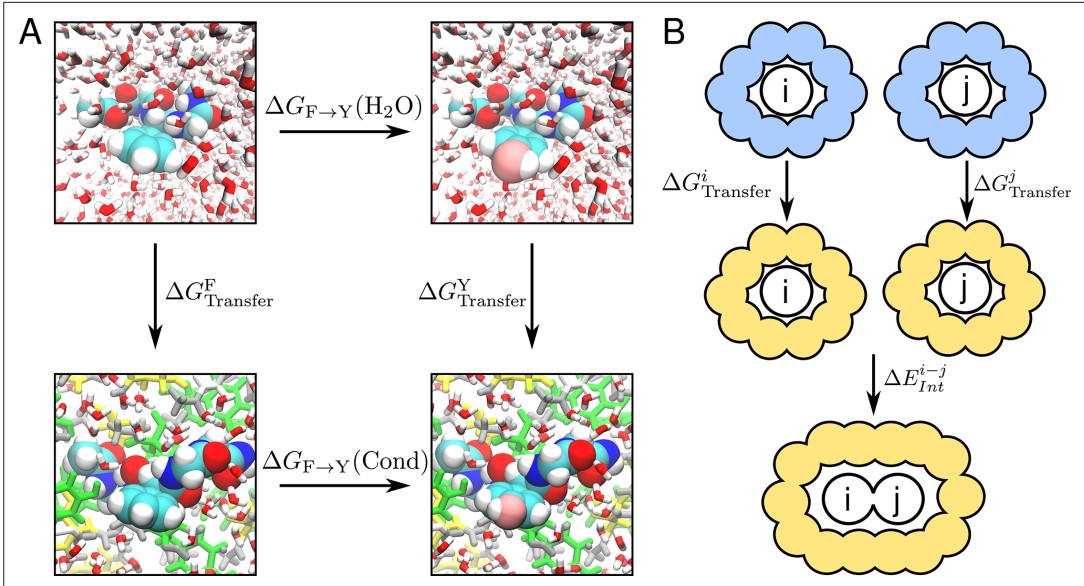

**Figure 2.** Computational approach to estimate differences in phase separation propensity between tyrosine and phenylalanine. (**A**) Thermodynamic cycle used in this study for the estimation of free energy differences upon mutation for the insertion of a peptide in a molecular condensate. (**B**) Schematic of the process of contact formation between two molecules $i$ and $j$ used in the quantum chemical calculations. We consider that contact formation involves the transfer from water (blue) to a different medium (orange) and the interaction in this medium between the entities involved.

most of our simulations, we have used the Amber ff99SB⋆-ILDN force field, including corrections for backbone and side chain torsions (***Best and Hummer, 2009***; ***Lindorff-Larsen et al., 2010***), and TIP3P water (***Jorgensen et al., 1983***). To evaluate the impact of the water model on our results, we have also prepared simulations with the TIP4P-Ew water model (***Horn et al., 2004***). For the simulations with different organic solvents, we used the GAFF parameters derived by Mobley and co-workers (***Bannan et al., 2016***). Tables with a comprehensive description of the simulation systems considered in this work can be found in ***Supplementary file 1***.

## Thermodynamic cycle

To estimate differences in the transfer free energy ($\Delta\Delta G_{\text{Transfer}}$) of our short peptides into a different medium —either a condensate or a different solvent— we use an alchemical (i.e. non-physical) transformation (***Mey et al., 2020***). Specifically, the thermodynamic cycle we construct involves four different states for the GGXGG peptide, with X=F ($\lambda = 0$) or X=Y ($\lambda = 1$), which is inserted either in water or in the alternative medium. We illustrate this thermodynamic cycle in the case of transfer into a condensate in ***Figure 2A***. The vertical branches of the thermodynamic cycle are associated with the free energy of transfer of the model peptides from water into the condensate ($\Delta G_{\text{Transfer}}^{\text{F}}$ and $\Delta G_{\text{Transfer}}^{\text{Y}}$), while the horizontal branches involve the alchemical transformation ($\Delta G_{\text{F}\rightarrow\text{Y}}(\text{H}_2\text{O})$ and $\Delta G_{\text{F}\rightarrow\text{Y}}(\text{Cond})$). Using simulations, we can easily calculate the free energies corresponding to the horizontal branches. This enables us to estimate the desired quantity as

$$\Delta\Delta G_{\text{Transfer}} = \Delta G_{\text{Transfer}}^{\text{Y}} - \Delta G_{\text{Transfer}}^{\text{F}} = \Delta G_{\text{F}\rightarrow\text{Y}}(\text{Cond}) - \Delta G_{\text{F}\rightarrow\text{Y}}(\text{H}_2\text{O}) \qquad (1)$$

Negative $\Delta\Delta G_{\text{Transfer}}$ values indicate that transferring the Tyr peptide into the condensate is more favourable than transferring the Phe peptide, whereas positive values indicate the opposite.

## Simulation setup

For the simulation runs in pure solvents, we simply inserted the GGXGG peptide in an octahedral box, leaving at least 1 nm of distance to each wall. Then the boxes were filled with the corresponding solvent molecules. For the simulations in the ternary model condensates, we first inserted the peptide

at the central position of a rectangular box with dimensions 8.5 nm × 4.5 nm × 4.5 nm, containing an equimolar mixture of Gly, Ser, and either Tyr or Phe dipeptides, which we term GSY and GSF, respectively. Then, the box was expanded up to 20 nm and filled with water molecules. This results in a simulation box where the peptide of interest is immersed in a protein-dense slab in contact with a dilute phase.

The equilibration protocol is identical in all cases. First, we ran an energy minimization using a steepest descent algorithm, followed by short simulations in the NVT and NPT ensembles, using the Berendsen (*Berendsen et al., 1984*) and velocity rescaling (*Bussi et al., 2007*) thermostats, respectively, to set the temperature at 298 K, and the Berendsen barostat (*Berendsen et al., 1984*) to set the pressure at 1 bar on the latter. Finally, we performed equilibrium simulations in the NPT ensemble using a leap-frog stochastic dynamics integrator with a time-step of 2 fs at 298 K and setting the pressure to 1 bar with the Parrinello-Rahman barostat (*Parrinello and Rahman, 1980*). The duration of the equilibrium runs was 1 $\mu$s for the runs with a condensate slab configuration and 500 ns for all other solvents (see *Supplementary file 1*). We ran triplicates of the GSY and GSF simulations. Note that for each system-solvent combination, two different sets of simulations had to be performed, one for each of the alchemical ($\lambda$) states corresponding to the central peptide residue being Phe ($\lambda = 0$) or Tyr ($\lambda = 1$).

For the alchemical transitions, we use a non-equilibrium switching method (*Seeliger and de Groot, 2010*; *Gapsys et al., 2015*; *Aldeghi et al., 2019*) often used for the calculation of differences in binding affinities (*Aldeghi et al., 2018*; *Gapsys et al., 2020*) or free energy changes upon mutation (*Gapsys et al., 2016*; *Aldeghi et al., 2019*; *Martinez-Martin et al., 2023*). After discarding the first 100–200 ns from the long equilibrium runs, we selected 100 evenly spaced snapshots as initial states. Then, we run short (50 ps) simulation trajectories where the $\lambda$-state was switched forward ($\lambda = 0 \rightarrow 1$) or backward ($\lambda = 1 \rightarrow 0$).

We have run additional simulations of the GSY and GSF condensates in the absence of the longer peptide using the temperature replica exchange method (REMD) (*Sugita and Okamoto, 1999*). For these simulations, we first set up systems using the protocol described above for the slab simulations. Then, we ran a 50 ns, high-temperature simulation at 500 K in the NVT ensemble. From this trajectory, we dumped 48 randomly selected snapshots that were used as initial conformations for the different replicas. We ran REMD for 500 ns at temperatures ranging between 293.5 and 411.6 K. Replica swaps were attempted every 2 ps. The first 200 ns for each replica were discarded from the analysis. To study interactions in the biomolecular condensates, we isolated the dense phase in the slab from the REMD replica at room temperature. This was then briefly re-equilibrated as before, and production simulations of the condensates were run for 1 $\mu$s.

For all simulations, we have used the Gromacs software package (*Abraham et al., 2015*) (v.2024). To generate hybrid topologies for the different $\lambda$-states, we have used the PMX software (*Seeliger and de Groot, 2010*; *Gapsys et al., 2015*; *Aldeghi et al., 2019*) (v. 3.0).

## Analysis

The simulations have been analyzed using a combination of analysis programs in the PMX and Gromacs packages (*Abraham et al., 2015*) and in-house Python scripts that make extensive use of the MDTraj library (*McGibbon et al., 2015*). For our free energy estimates, we have analyzed the non-equilibrium switching simulations using the PMX tools (*Seeliger and de Groot, 2010*; *Gapsys et al., 2015*; *Aldeghi et al., 2019*). Specifically, we calculate the work values from the forward and backward transitions (i.e. $W = \int_{\lambda=0}^{\lambda=1} \frac{\partial H}{\partial \lambda} d\lambda$) and then estimate free energy differences between the initial and final states using Bennett's Acceptance Ratio (*Bennett, 1976*). For more details, we refer to the PMX papers (*Seeliger and de Groot, 2010*; *Gapsys et al., 2015*; *Aldeghi et al., 2019*). Errors were estimated from bootstrapping, except in the case of the GSY and GSF condensates, where we estimate the standard error of the mean from the three replicates.

To obtain the phase diagrams, we calculate the peptide densities of the dilute and dense phases ($\rho_L$ and $\rho_H$, respectively) from the REMD trajectories. These were obtained after fitting the density profile to the expression

$$\rho(x) = \frac{\rho_L + \rho_H}{2} + \frac{\rho_L - \rho_H}{2} \times \tanh\left(\frac{|x| - x_{DS}}{t}\right) \tag{2}$$

where $x_{DS}$ and $t$ are, respectively, the position and width of the dividing surface between dense and dilute phases (*Tesei et al., 2021*). Then, using data only for temperatures where we could clearly distinguish between dense and dilute phases, we obtained the critical temperature ($T_c$) from a fit to the equation

$$\rho_H - \rho_L = A(T_c - T)^\beta \tag{3}$$

where $\beta = 0.325$ is the critical exponent (*Dignon et al., 2018*). The critical density ($\rho_c$) was derived from the law of rectilinear diameters

$$\frac{\rho_H + \rho_L}{2} = \rho_c + c(T_c - T). \tag{4}$$

We used gmx dipoles to calculate the dielectric constant ($\varepsilon$) for different condensates and solvents from the fluctuations in the dipole moment of the simulation box, $\mathbf{M}$, which is defined as (*Neumann, 1983*)

$$\varepsilon = 1 + \frac{\langle \mathbf{M}^2 \rangle - \langle \mathbf{M} \rangle^2}{3\varepsilon_0 k_B T \langle V \rangle}. \tag{5}$$

In this expression, $\varepsilon_0$ is the vacuum permittivity, $k_B$ is Boltzmann's constant, and $V$ is the volume of the simulation box. For the calculation of the dielectric constant of condensates, we used the simulations of isolated dense phases mentioned above. Error analysis has been performed by blocking or determined as the standard error of the mean when derived from multiple trajectories.

## Quantum chemical calculations

We performed quantum mechanical calculations using the Gaussian 16 program to study the interaction energy of the different complexes. All calculations were made at the Density Functional Theory (DFT) level, using the $\omega$B97XD functional (*Chai and Head-Gordon, 2008*) and Pople's 6–311++G(d,p) basis set (*Hehre et al., 1972*) for geometry optimization. Energetics were refined with the 6–311++G(3df,2p) basis set. Geometry optimizations were done in different solvents, using the Polarizable Continuum Model approach (*Mennucci, 2012*). The interaction energy in a given solvent was calculated according to

$$\Delta E_{\text{Int}}^{ij} = E^{ij} - \left( E^i + E^j \right) \tag{6}$$

where $i$ and $j$ specify the interacting monomers. We consider p-cresol to represent the side chain of Tyr (Y hereafter), toluene to represent the side chain of Phe (F hereafter), and N-methyl-acetamide to represent an amide bond group (A hereafter). Dimers $ij$ include different combinations of Y, F, and A in various orientations.

To calculate the free energy transfer of a monomer from water to a given solvent, we used the SMD solvation model (*Marenich et al., 2009*) to calculate accurate solvation-free energies of the monomers in different solvents, using B3LYP/6–31+G(d,p) level of theory (*Becke, 1993*; *Lee et al., 1988*). Thus, the transfer free energy from water to a given solvent $s$ is calculated according to

$$\Delta G_{\text{Transfer}}^i = \Delta G^i(s) - \Delta G^i(\text{H}_2\text{O}) \tag{7}$$

As solvents, we have considered three different groups:

1. Alcohols: methanol, ethanol, 1-propanol, 1-butanol, 1-pentanol, 1-hexanol, 1-nonanol, and 1-decanol.
2. Aliphatic solvents: 2-nitropropane, 2-bromopropane, 1-bromopropane, 1-bromopentane, 1-fluorooctane, n-pentane, n-decane, nitrobenzene, o-dichlorobenzene, bromobenzene, iodobenzene, ethylbenzene, benzene, toluene, and cyclohexane.
3. Water with varying dielectric values.

These solvents span a broad range of values of the dielectric constant and can be viewed to represent different chemical environments where contact formation can take place.

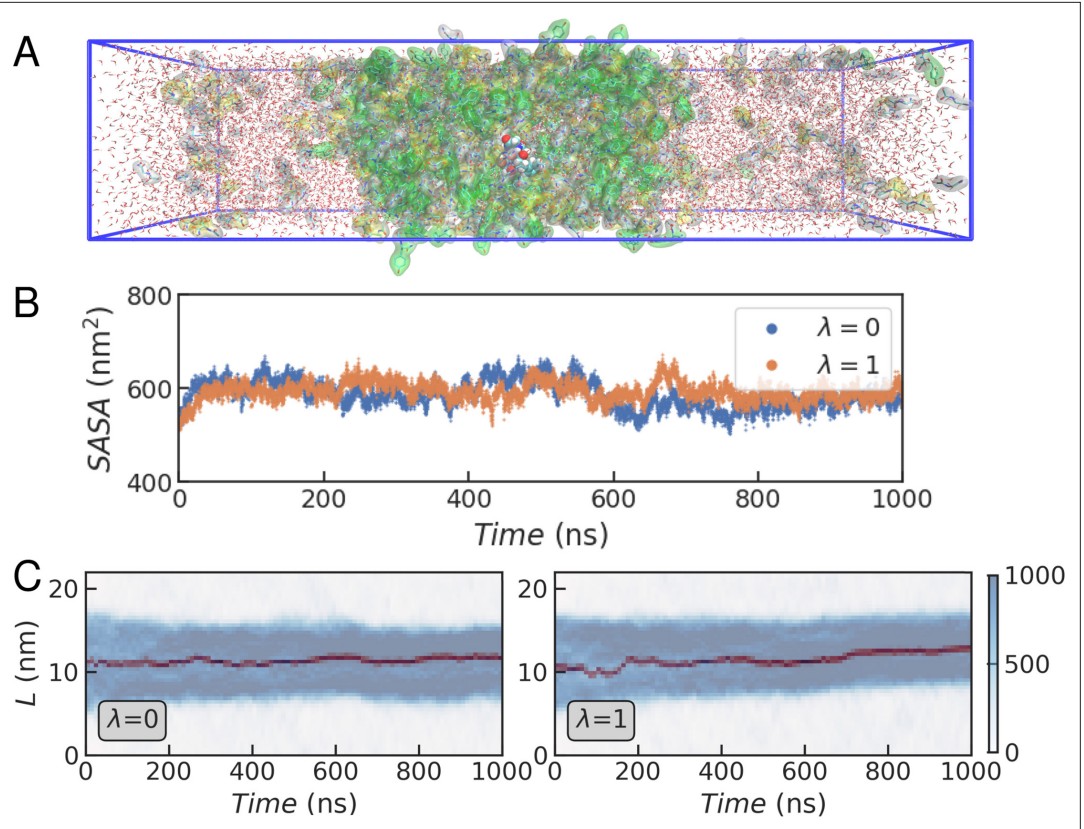

**Figure 3.** Molecular simulations of the GG(F/Y)GG peptide in a GSY peptide slab. (**A**) Representative simulation box with a fully solvated GSY condensate in slab geometry, including a GGXGG peptide (spheres) and the capped amino acid mixture (G: white, S: yellow, and Y: green). (**B**) Time series for the solvent accessible surface area (*SASA*) in a representative trajectory of the GGXGG peptide within the GSY condensate for different values of $\lambda$. (**C**) Time evolution of the density profiles calculated across the longest dimension of the simulation box (*L*) in the coexistence simulations. In blue, we show the density in mg/mL of all the peptides, and in dark red that of the F/Y residue in the GGXGG peptide.

The online version of this article includes the following figure supplement(s) for figure 3:

**Figure supplement 1.** Replicates of the GGXGG peptide simulations in the GSY condensate.

**Figure supplement 2.** Work values for the forward and reverse alchemical transitions for different initial snapshots (bottom) and their distributions (top).

**Figure supplement 3.** Results for the GSY condensate in the TIP4P-Ew water model.

**Figure supplement 4.** Results for the GSY condensate in the TIP4P-Ew water model.

**Figure supplement 5.** Replicates of the GGXGG peptide simulations in the GSF condensate.

## Results and discussion

### Transfer free energies into a condensate slab

Our first goal is to find whether differences in transfer free energy for Phe and Tyr variants capture the hierarchy in sticker strength derived from experiments. Ideally, one would calculate the transfer free energies for the protein of interest inside a realistic protein condensate. However, this would involve simulations of multiple copies of an IDR with tens or hundreds of amino acids, resulting in a system with very slow conformational dynamics (*Paloni et al., 2020*; *Zheng et al., 2020*; *Galvanetto et al., 2023*). As a computationally efficient alternative, we use a simple model peptide with sequence GGXGG, with X being either F or Y, which is transferred into the dense mixtures of short peptides we have recently characterized (*De Sancho, 2022*) (see *Figure 3A*). Peptide condensates are useful model systems as they retain many properties of protein condensates, but allow for much

greater computational efficiency (*Paloni et al., 2020*; *Tang et al., 2021*; *De Sancho, 2022*; *Brown and Potoyan, 2024*).

We define the thermodynamic cycle in *Figure 2* and estimate free energies associated with its horizontal branches using alchemical transformations, which we performed using a state-of-the-art non-equilibrium switching method (*Seeliger and de Groot, 2010*; *Gapsys et al., 2015*; *Aldeghi et al., 2019*) (see Methods). The first step in this procedure involves running long equilibrium simulations of the peptide of interest at the initial $\lambda$-states (i.e. where X=F for $\lambda = 0$ and X=Y for $\lambda = 1$). While these calculations are straightforward for a peptide in solution, within the condensate, we must first ensure that the condensate slab has converged to a stable density. Also, the peptide that experiences the transformation, which is not restrained, must remain buried within the condensate for all the snapshots that we use as initial frames, to avoid averaging the work in the dilute and dense phases. In *Figure 3B*, we show the solvent-accessible surface area (*SASA*) of all protein residues in a representative trajectory of the GSY condensate with the model peptide submerged. After an initial increase in *SASA*, the values stabilise and remain flat for the remainder of the simulation, which is indicative of stable dense and dilute phases. We also show the density profiles for all the amino acid residues in the condensate and for the mutated residue (*Figure 3C*). For both $\lambda$ values, we find that during the whole duration of the simulation, the GGXGG peptide remains buried within the condensate. Very similar results were obtained in three independent replicates started from different initial configurations of the peptide slab (see *Figure 3—figure supplement 1*).

We discarded the initial 100 ns of these long trajectories and selected initial configurations for non-equilibrium switching from evenly spaced snapshots of the $\lambda = 0$ and $\lambda = 1$ datasets. For each of the replicates, we ran 100 independent simulations forward and backwards, collecting the values of $(\partial H/\partial \lambda)$. Alchemical transformations were also performed in the same way for the peptide in TIP3P water (see Methods). Then, we estimate transfer free energy differences by combining data from the transformations within the condensate and in water (see *Figure 3—figure supplement 2A and B*). We find that $\Delta\Delta G_{\text{Transfer}}$ is small and negative (-2.9 ± 0.5 kJ/mol), indicating that transfer into the peptide condensate consisting of Gly, Ser, and Tyr is more favourable for the Tyr-containing peptide than for the Phe variant. This result is consistent with mounting experimental evidence that suggests a greater propensity for Tyr to form condensates compared to Phe in variants of proteins such as FUS, LAF1, and hnRNPA1.

A limitation of our model peptide condensates is their high protein densities, which are larger than those of full-length IDRs (*Zheng et al., 2020*). This may be influenced by the propensity of the TIP3P water model to induce compact states (*Piana et al., 2015*). We have repeated the same procedure using the TIP4P-Ew water model (*Horn et al., 2004*) to estimate $\Delta\Delta G_{\text{Transfer}}$ for the GGXGG peptide into a GSY condensate (see *Figure 3—figure supplement 3*). In combination with force fields from the Amber ff99SB family, this water model resulted in improved properties for peptides in dilute (*Nerenberg and Head-Gordon, 2011*) and dense peptide solutions (*Miller et al., 2016*). With this force field-water model combination, we have obtained a value of -2.6 ± 0.6 kJ/mol (see *Figure 3—figure supplement 4*), which is in good agreement with the result in TIP3P water.

One possible explanation for these results is the formation of preferential sticker-sticker interactions between the GGYGG peptide and the Tyr residues in the GSY condensate. While the hydroxyl group of Tyr can form hydrogen bonds, such interactions are expected to be depleted for the GGFGG variant. Conversely, if the condensate were composed of Phe instead of Tyr, there would be no reason to favour GGYGG. Moreover, both hydrophobicity considerations and interaction energies from contact matrices suggest that transfer into a GSF condensate should be more favourable for the GGFGG peptide than for GGYGG. To test this hypothesis, we have run the same procedure in a Gly/Ser/Phe (GSF) peptide condensate. We show the time series data for *SASA* and protein density for the GSF simulations in *Figure 3—figure supplement 5*. Combining these results with those from the simulations in water, we obtain a value for the transfer free energy difference of $\Delta\Delta G_{\text{Transfer}} =$ -2.5 ± 0.5 kJ/mol (see *Figure 3—figure supplement 2*). Hence, irrespective of the aromatic amino acid within the condensate, transfer is more favourable for the Tyr-containing peptide.

## Similar protein-protein interaction patterns in Tyr and Phe condensates

To gain further insight into the dominant interactions, we have run additional simulations of isolated dense phases for GSY and GSF condensates, without the GGXGG peptide (see *Figure 4A*). First,

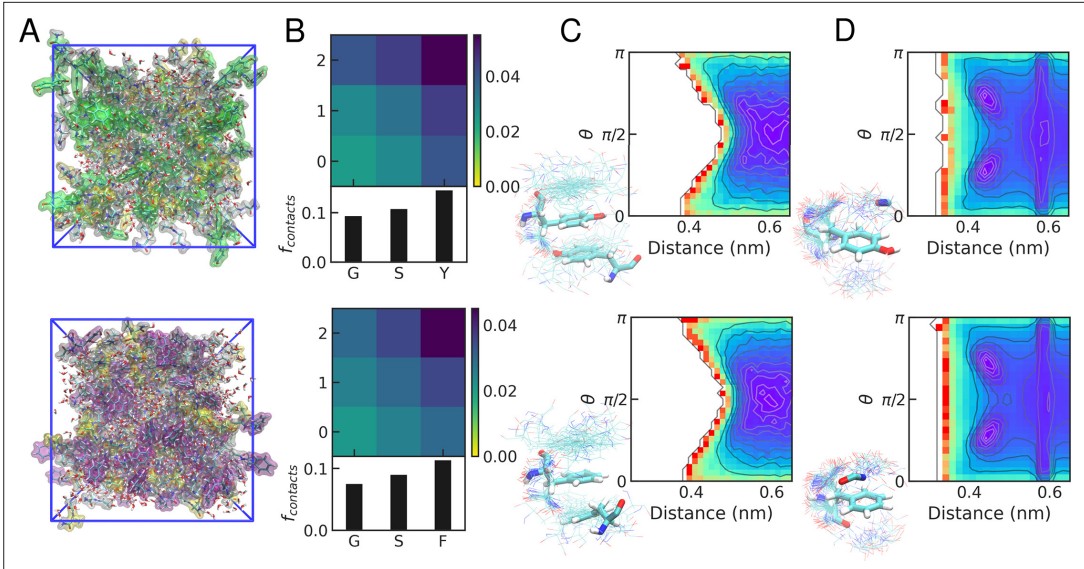

**Figure 4.** Interaction patterns in dense phases of peptide condensates. (**A**) Representative snapshots from the simulations of dense phases. G/S/F/Y dipeptides are shown as transparent surfaces (G: white, S: yellow, Y: green, F: purple). (**B**) Interaction matrix for the normalized number of contacts between different pairs of amino acid residues in the condensates (top) and for each type of amino acid (bottom). (**C**) Density plots for side chain interactions between aromatic side chains, as characterized by the mean inter-residue distance and the angle $\theta$ between the vectors normal to the rings (**Calinsky and Levy, 2024**). (**D**) Density plots for sp$^2$-$\pi$ interactions between amide bonds and aromatic side chains, as characterized by the mean inter-group distance and the angle $\theta$ between the vectors normal to the peptide bond and ring planes. In all panels, results for GSY and GSF condensates are shown on the top and bottom, respectively. Representative snapshots of relevant interactions for each type of pair are shown.

we look at the statistics for the different interaction pairs inside the condensates. We find that the interaction patterns are very similar in both cases, with a greater number of contacts formed by the aromatic stickers and smaller contributions from Gly and Ser (see *Figure 4B*). This result recapitulates our previous observations (*De Sancho, 2022*) and also those from atomistic simulations of low complexity regions of FUS and Ddx4 (*Zheng et al., 2020*). The greater contact probability of Tyr/Phe relative to Gly/Ser is consistent with the scaffold-client relationship between aromatic and polar residues (*Dignon et al., 2020*).

To focus on $\pi - \pi$ interactions between pairs of aromatic side chains, we have also calculated the mean inter-side chain distances and the angles between vectors normal to the aromatic rings ($\theta$). In *Figure 4*, we show their distribution, which is very similar in the GSY and GSF condensates. Even though planar $\pi - \pi$ stacking is feasible, it does not play a dominant role (see the regions with low inter-residue distance and $\theta$ angles close to 0 or $\pi$). We also examine sp$^2$-$\pi$ contacts formed between aromatic and peptide bond groups, which are common in our peptide dense phases. This is consistent with previous simulation work (*Zheng et al., 2020*; *Murthy et al., 2021*; *Rekhi et al., 2024*) and also with a statistical analysis of experimental structures from the PDB, which showed that these interactions are particularly prevalent in disordered regions of folded proteins (*Vernon et al., 2018*). Interestingly, also for these sp$^2$-$\pi$ interactions, we find no significant differences in interaction patterns between GSY and GSF condensates.

## Same interaction patterns, different stabilities

Our results indicate that Tyr's greater ability to promote condensate formation, relative to Phe, is not primarily due to differences in protein–protein interaction patterns between GSY and GSF condensates. Importantly, this similarity does not imply that the two systems have an equal propensity for phase separation, as the energetics of the interactions may differ. To interrogate this point, we have calculated the phase diagrams for the GSY and GSF condensates. We have used REMD simulations to estimate the peptide density of the dilute and dense phases at the temperatures where these

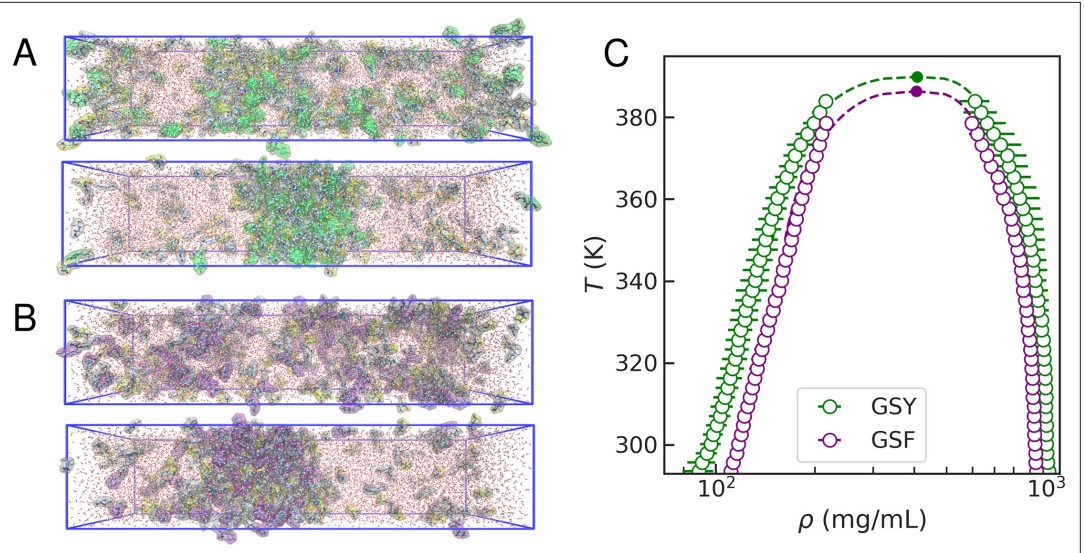

**Figure 5.** Estimation of phase diagrams from atomistic simulations. (**A**) Representative snapshots of replica exchange method (REMD) simulations of the GSY condensate at high (top) and low (bottom) temperatures. Color code as in **Figure 3**. (**B**) Same for the GSF condensate. (**C**) Phase diagrams for GSY (green) and GSF (purple). Empty circles correspond to simulations and filled circles correspond to fitted critical temperatures ($T_c$) and densities ($\rho_c$).

The online version of this article includes the following figure supplement(s) for figure 5:

**Figure supplement 1.** Density profiles at different temperatures for the GSY (**A**) and GSF (**B**) condensates.

**Figure supplement 2.** Density profiles for sticker residues across the simulation box from the replica exchange method (REMD) simulations at room temperature.

can be resolved (see snapshots at high and low temperatures in **Figure 5A–B** and density profiles in **Figure 5—figure supplement 1**). In **Figure 5C**, we show the calculated phase diagrams for both systems, which have remarkably similar shapes. However, the saturation density ($\rho_{sat}$), i.e., the amount of peptide required for the system to phase separate, is slightly lower in the GSY condensate than in GSF. Additionally, GSY has a slightly greater critical temperature ($T_c$).

For simplicity, we have calculated the densities of all the peptides in the mixture as if they were partitioning uniformly. However, a more rigorous analysis would involve treating stickers and spacers separately, as we did in our systematic study for mixtures with different concentrations of stickers and spacers (**De Sancho, 2022**). Given the role as a 'scaffold' of the aromatics in the dense phases, we expect them to be overrepresented in the condensate relative to the spacers ('clients'). In **Figure 5— figure supplement 2**, we show the density profiles of aromatics at room temperature, where we find that the saturation density of Tyr is lower than that of Phe. This overemphasises that, even if Phe and Tyr play similar roles as scaffolds in peptide condensates, Tyr has a stronger propensity to phase separate, in agreement with the hierarchy of sticker strengths observed experimentally.

## Condensates as hydrated peptide solvents

We have thus far found that condensate formation by Tyr and Phe is mediated by similar contact patterns and interaction modes, dominated by contacts between aromatic residue pairs, which are more favourable for Tyr. An important driver of the observed behaviour may involve differences in protein-solvent interactions by the Phe/Tyr peptides. To investigate this, we begin by analysing the distribution of water molecules around the aromatic side chains in the dense phase simulations of GSF and GSY (see **Figure 4A**), revealing notable differences. In **Figure 6A**, we show the radial distribution function ($g(r)$) for water oxygen around the $C^\zeta$ in the aromatic side chains. In the case of Tyr, there is a prominent peak in $g(r)$ at short distances due to the presence of water molecules involved in hydrogen bonds. These interactions are enthalpically favourable and entropically unfavourable due to the water structuring around the Tyr hydroxyl group (**Lin et al., 2017**). The different thermodynamic contributions are intertwined in the calculated $\Delta\Delta G_{Transfer}$. The net balance will depend on whether

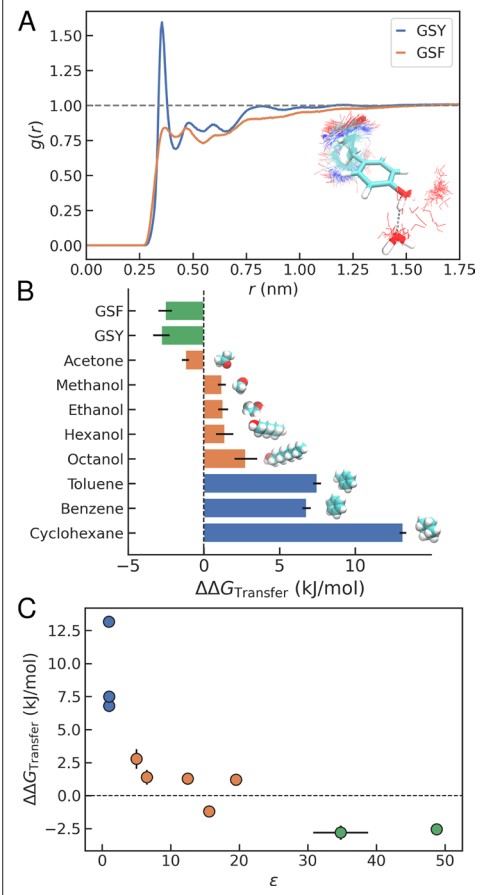

**Figure 6.** Comparison of peptide condensates with organic solvents. (**A**) Radial distribution function for water oxygen around the $C^\zeta$ in Phe/Tyr for GSF and GSY condensates. We show a representative overlay of simulation snapshots where water molecules are hydrogen-bonded to the Tyr side chain. (**B**) Transfer free energy differences from water to a different medium between Tyr and Phe. We consider condensates (green), polar solvents (orange) and apolar solvents (blue). (**C**) Same as a function of calculated dielectric constant, $\varepsilon$.

The online version of this article includes the following figure supplement(s) for figure 6:

**Figure supplement 1.** Work values for the forward and reverse alchemical transitions for different initial snapshots (left) and their distributions (right).

the peptide is transferred from water into a condensate, into the core of a folded protein or an altogether different solvent, capable or not of hydrogen bonding.

The picture that emerges from our transfer free energy calculations is one where condensates act as media capable of interacting with the peptide via different types of interactions, including $\pi - \pi$ and $sp^2$-$\pi$ contacts. To investigate the effect of different environments, we have performed the same type of calculations using a battery of solvents with diverse properties. These include cyclohexane or octanol —used in the past as models for the interior of folded proteins (*Kauzmann, 1959*; *Nozaki and Tanford, 1971*; *Radzicka and Wolfenden, 1988*; *Wimley et al., 1996*; *Pace et al., 2011*)— and also shorter alcohols or acetone. The results of these calculations are shown, together with those of the different peptide condensates and solutions, in *Figure 6B* (see also *Figure 6— figure supplement 1*). In the case of a purely non-polar solvent like cyclohexane, the difference in free energy from Phe to Tyr is large and positive ($\Delta\Delta G_{\text{Transfer}} = 13.2 \pm 0.3$ kJ/mol), indicating that Phe is favoured. This is also the case for aromatic solvents like benzene and toluene (with $\Delta\Delta G_{\text{Transfer}}$ of $6.8 \pm 0.3$ and $7.5 \pm 0.3$ kJ/mol, respectively). Lacking partners for the interactions that favour Tyr in the target solvent, the greater hydrophobicity of Phe dominates in the net $\Delta\Delta G_{\text{Transfer}}$. This is consistent with the stronger interaction strength of Phe-Phe interactions in the Miyazawa and Jernigan potential (*Miyazawa and Jernigan, 1996*) (see *Figure 1C*), derived from contacts in the apolar cores of proteins. The difference in free energy is smaller for alcohols and decreases gradually as the length of the aliphatic chain decreases. For acetone, transfer of Tyr is slightly more favourable ($\Delta\Delta G_{\text{Transfer}} = -1.2 \pm 0.2$ kJ/mol).

A parameter that we can determine from our MD trajectories characterising the properties of the different solvents, and also the condensates, is the dielectric constant ($\varepsilon$, see Methods). Clearly, the $\Delta\Delta G_{\text{Transfer}}$ values have a dependence on the value of $\varepsilon$ (see *Figure 6C*). For the GSY and GSF condensates, $\varepsilon$ values range between 30 and 50, surprisingly close to that derived from experiments on Ddx condensates using Flory-Huggins theory ($45 \pm 13$) (*Nott et al., 2015*) and from atomistic simulations of Ddx4 (~35-50 at a volume fraction of $\phi = 0.4$) (*Das et al., 2020*). In summary, for non-polar solvents with low $\varepsilon$, Phe is preferred to Tyr, while for polar solvents, including the condensates, the trend is reversed. At intermediate values of $\varepsilon$, there is a crossover between a regime dominated by hydrophobicity and one favoured by the stronger interactions established by Tyr.

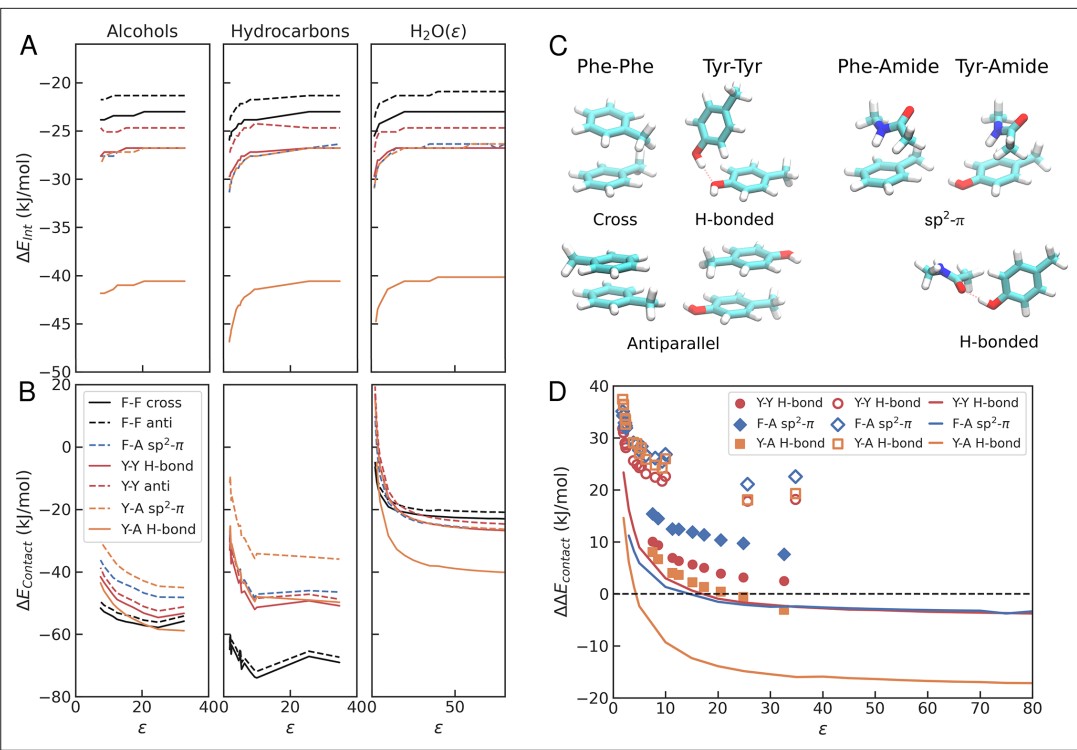

**Figure 7.** Contact energies from quantum chemical calculations. (**A**) $\Delta E_{Int}^{ij}$ and (**B**) $\Delta E_{contact}^{ij}$ for different interaction pairs $ij$ and different solvents as a function of the dielectric constant, $\varepsilon$. (**C**) Optimized geometries from quantum chemical calculations for Phe and Tyr interaction pairs. (**D**) $\Delta\Delta E_{Contact}^{Solvent}$ with respect to the cross-F-F interaction for different solvents as a function of the dielectric. Different symbols correspond to the different solvent types. Filled: alcohols; empty: alkanes/benzenes; lines: water with different dielectric.

## Quantum calculations confirm a crossover in interaction strength

To characterise these subtle effects in greater detail, we resort to quantum chemical calculations, which can more accurately determine interaction strengths for specific configurations involving aromatic residues (*Carter-Fenk et al., 2023*; *Calinsky and Levy, 2024*). We consider different modes of interaction between the side chains of Phe and Tyr, represented by toluene (F) and p-cresol (Y), respectively. We considered various orientations in the optimization process and selected the two most stable structures for each case. For F-F, the most stable structures correspond to cross and antiparallel configurations, whereas in the case of Y-Y, H-bonded and antiparallel are most favoured (see *Figure 7A and C*). Our results are consistent with previous work on the toluene dimer, where the energy minima also corresponded to the cross and antiparallel structures, with only minor differences in their energies (*Tsuzuki et al., 2005*; *Rogers et al., 2006*). Other structures, like parallel or T-shaped, are less stable for toluene, although in the benzene dimer, T-shaped conformers can be favoured (*Chipot et al., 1996*). In our optimizations, T-shaped F-F structures tended to collapse to less stable parallel structures, and hence we do not include them here. The interaction energy values we obtained are qualitatively similar to those in the literature (*Calinsky and Levy, 2024*), though slightly more negative, and they follow the same rank order (i.e. Y-Y H-bonded < antiparallel Y-Y < cross F-F). We also calculate the energies for Phe and Tyr side chains interacting with an amide group (A) representing a peptide bond in the prevalent sp²-π interactions (*Vernon et al., 2018*) (see *Figure 7C*). In the case of Tyr, we have also analyzed the configuration where it forms a hydrogen bond with the amide through the alcohol group of p-cresol.

To explore the environmental dependence in contact formation for amino acid pairs, we have calculated values of the interaction energies ($\Delta E_{Int}^{ij}$) considering a set of protic and non-protic solvents (see *Figure 7A*). Additionally, we have run the same calculations using a modified water with the dielectric constant as a free parameter. The overall trends in interaction energies are very similar across interaction types. In all cases, the interaction energy tends to become stronger as the value of the dielectric

decreases. Also, the relative strength of interactions is consistent across solvents. The strongest interaction corresponds to the Y-A hydrogen-bonded interaction. Next comes a series of partial $\pi$-stacking configurations with contributions of hydrogen bonds and sp2-$\pi$ interactions, namely, Y-Y H-bond, Y-A sp$^2$-$\pi$, and F-A sp$^2$-$\pi$ structures, which we have found to be particularly abundant in our MD runs. Finally, the $\pi$-stacked F-F cross is the weakest interaction among those included in this analysis.

Therefore, if we consider these interaction energies as the primary driving force for condensate formation, Tyr should be more adhesive than Phe independently of the environment. However, as realized by Miyazawa and Jernigan (*Miyazawa and Jernigan, 1996*; *Thomas and Dill, 1996*), contact formation, either within the core of folded proteins or in the case of molecular condensates, involves two processes: first, the transfer of these groups from water to an environment with a lower dielectric; and second, the interaction between side chain groups (or with the backbone), which is captured by $\Delta E^{ij}_{\mathrm{Int}}$ (see *Figure 2B*). Hence, for a given solvent, the total energy can be calculated from the sum of these two contributions as

$$\Delta E^{ij}_{\mathrm{Contact}} = \Delta E^{ij}_{\mathrm{Int}} + \Delta G^{i}_{\mathrm{Transfer}} + \Delta G^{j}_{\mathrm{Transfer}} \tag{8}$$

The last two terms on the right-hand side of this expression capture the propensity of a monomer to be transferred from water to the solvent, which can be modelled using polarizable continuum models such as SMD (*Marenich et al., 2009*). Notice that when the solvent is a low dielectric nonprotic solvent such as cyclohexane, $\Delta G^{i}_{\mathrm{Transfer}}$ recapitulates the results from a standard hydrophobicity scale (*Kyte and Doolittle, 1982*).

When both contributions are considered, the picture changes dramatically (see *Figure 7B*). In nonprotic aliphatic solvents, such as alkanes and benzene derivatives, interaction for F-F pairs becomes more favourable than in Y-Y pairs, irrespective of the dielectric constant. On the other hand, in alcohols, $\Delta E^{ij}_{\mathrm{Contact}}$ tends to favour Y-A interactions at high $\varepsilon$ and F-F pairs at low $\varepsilon$. In other words, at intermediate dielectrics, we find a crossover between F and Y in the propensity to form pairwise contacts in solvents. In our calculations, we have also considered a water-like solvent where the dielectric constant can be tuned as a free parameter (see *Figure 7C*). Also, we find a similar tendency in this case, but now the crossover is obtained at slightly lower values of $\varepsilon$. In *Figure 7D*, we show $\Delta\Delta E^{ij}_{\mathrm{Contact}}$ values, where we are using as reference the energy of the F-F cross configuration, for three selected interactions: Y-Y H-bond, Y-A H-bond, and F-A, for the three types of environments. Negative values of $\Delta\Delta E^{ij}_{\mathrm{Contact}}$ indicate a preference of the corresponding interaction relative to F-F cross, while positive values indicate that cross-F-F is stronger. Clearly, there is a crossover between a regime where toluene forms stronger interactions and one where p-cresol is preferred, which is highly influenced by the type of environment. In a non-protic solvent, where hydrogen bonding is not possible, or in a protic solvent with low dielectric constant, we find that F-F interactions are favoured. In contrast, in high dielectric environments, Y-Y and Y-A interactions dominate.

## Conclusions

In this work, we have aimed to reconcile the apparent contradictions between the lower solvation-free energy (*Wolfenden et al., 1981*; *Chang et al., 2007*) and stronger hydrophobicity of Phe (*Kyte and Doolittle, 1982*; *Tesei et al., 2021*) and the greater contact energies assigned to Phe-Phe interactions in proteins (*Miyazawa and Jernigan, 1996*) with the higher sticker strength of Tyr in biomolecular condensates (*Lin et al., 2017*; *Wang et al., 2018*; *Schuster et al., 2020*; *Bremer et al., 2022*). In the past, the different propensities of positively charged stickers to form condensates had been explained from the fundamental properties of Lys and Arg (*Wang et al., 2018*; *Das et al., 2020*; *Gobbi and Frenking, 1993*; *Hong et al., 2022*). However, an explanation for the different strengths of Phe and Tyr remained elusive (*Das et al., 2020*).

Through a combination of classical MD simulations and quantum mechanical calculations, we have found that, in addition to the distinct types of interactions formed by Phe and Tyr, a crucial factor in their overall propensity for contact formation across different environments is the transfer free energy. At one extreme, we find environments such as the well-packed core of a protein, where the dielectric constant can be as low as 2–4 (*Pitera et al., 2001*). This type of environment can be modelled using aliphatic solvents, an approach followed extensively in the past (*Kauzmann, 1959*; *Nozaki and Tanford, 1971*; *Radzicka and Wolfenden, 1988*; *Wimley et al., 1996*; *Pace et al., 2011*). In this low-dielectric regime, Phe-Phe interactions are the most favourable,

as evidenced by both our classical transfer free energies (*Figure 6C*) and the estimated quantum $\Delta\Delta E^{ij}_{\text{Contact}}$ values (*Figure 7*). This explains why Phe forms the strongest interactions in contact matrices like that by Miyazawa and Jernigan, which were derived from statistics from 3D structures of proteins (*Miyazawa and Jernigan, 1996*) and are known to be dominated by hydrophobicity effects (*Pokarowski et al., 2005*). At the other extreme, in environments with high dielectric constants, such as aqueous or polar solvent conditions, Tyr-Tyr interactions become more favourable, as inferred from potentials of mean force derived from atomistic molecular simulations (*Chelli et al., 2002*; *Joseph et al., 2021*). Biomolecular condensates, due to their high water content (~200–300 mg/mL in our simulations, but up to ~600 mg/mL in experiments, *Zheng et al., 2020*), are best modelled as a solvent with an intermediate dielectric and favourable interactions between stickers. Hence, they operate in the second regime, although the influence of the dielectric is subtle, due to the possibility of crossover. We note that, although we have not included in our analysis positively charged residues that form cation-$\pi$ interactions with aromatics, the observed crossover will also be relevant to Arg/Lys contacts with Phe and Tyr. Following the rationale of our findings, within condensates, cation-Tyr interactions are expected to promote phase separation more strongly than cation-Phe pairs. Our findings align with a recent report from Rekhi et al. using the equilibrium alchemical techniques that emphasizes the role of transfer free energy in biomolecular condensates (*Rekhi and Mittal, 2025*).

The conclusions from our work are necessarily limited due to our reductionist approach. First, in our calculations, we are considering very short peptides including a single sticker, and hence we cannot account for the constraints imposed by chain connectivity (*Shimizu and Chan, 2001*) or the influence of multivalency in protein IDRs (*Martin et al., 2020*). Also, we have used a minimal alphabet of possible interaction pairs limited to two types of amino acid residues. We do not consider effects relative to cation-$\pi$ interactions that have been characterised by others (*Brady et al., 2017*; *Das et al., 2020*; *Aldeghi et al., 2018*; *Hazra and Levy, 2023*; *Calinsky and Levy, 2024*). In our calculations, the minimal peptides result in high protein densities in the dense phases, which are nonetheless consistent with those found by other authors in peptides with greater variation in length and sequence space (*Workman et al., 2024*; *Brown and Potoyan, 2024*; *Emelianova et al., 2025*). Additionally, results may vary somewhat when considering different classical force fields. Recent work points to a substantial variation in hydrophobicity across force fields (*Lobo et al., 2025*). Another potential limitation may arise from the choice of water model, as the dielectric constant —which plays an important role in our observations— is known to vary significantly between water models. In the case of the TIP3P model used in most of our calculations, it exceeds the experimental value ($\varepsilon = 100 \pm 5$, *Braun et al., 2014*). However, using the TIP4P-Ew water model (with $\varepsilon = 64 \pm 1$, *Horn et al., 2004*), we have found consistent results. Careful benchmarking of force fields against properties of peptide condensates offers a promising avenue for future efforts in force-field calibration.

The methodology proposed in the present work can be easily extended to other types of interactions. The overall qualitative conclusions on the crossover in interaction strengths as a function of the environment are likely to be maintained if different quantum approaches are used for contact calculations or different force fields are used for the classical simulations. The context dependence of interaction strengths that we observe is related to a growing body of work in the study of peptide self-assembly (*Kaygisiz et al., 2025*). The effect has deep roots in the concepts of 'pair' and 'bulk' hydrophobicity (*Wood and Thompson, 1990*; *Shimizu and Chan, 2002*). While pair hydrophobicity is connected to dimerization equilibria (i.e. the second step in *Figure 2B*), bulk hydrophobicity is related to transfer processes (the first step). Our work stresses the importance of considering both the pair contribution that dominates at high solvation, and the transfer free energy contribution, which overwhelms the interaction strength at low dielectrics.

## Acknowledgements

The authors gratefully acknowledge conversations with Priya R Banerjee, Rohit Pappu, and Tanja Mittag, which inspired this research. The work has been financed by grant PID2021-127907NB-I00 funded by MCIN/AEI/10.13039/501100011033, and the Basque Government (Project IT1584-22). The authors also thank the IZO-SGI SGIker (UPV/EHU/ERDF, EU) and DIPC for technical and human support and for the allocation of computational resources. The authors thankfully acknowledge RES resources provided by the Barcelona Supercomputing Center in MareNostrum5 (BCV-2025-2-0002).

# Additional information

## Funding

| Funder | Grant reference number | Author |
|---|---|---|
| MCIN/AEI | PID2021-127907NB-I00 | David De Sancho<br>Xabier Lopez |
| Basque Government | Project IT1584-22 | David De Sancho<br>Xabier Lopez |

The funders had no role in study design, data collection and interpretation, or the decision to submit the work for publication.

## Author contributions

David De Sancho, Xabier Lopez, Conceptualization, Formal analysis, Investigation, Methodology, Writing – original draft, Writing – review and editing

## Author ORCIDs

David De Sancho ⓘ https://orcid.org/0000-0002-8985-2685
Xabier Lopez ⓘ https://orcid.org/0000-0002-2711-3588

Reviewer #1 (Public review): https://doi.org/10.7554/eLife.104950.3.sa1
Reviewer #2 (Public review): https://doi.org/10.7554/eLife.104950.3.sa2
Author response https://doi.org/10.7554/eLife.104950.3.sa3

---

# Additional files

## Supplementary files

MDAR checklist

Supplementary file 1. Supporting Tables with additional details of the molecular simulations.

## Data availability

Additional simulation input files and analysis scripts are openly accessible at https://github.com/BioKT/Stickers/tree/master/PHETYR (copy archived at *De Sancho, 2025*).

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
