## [Editor Report · eLife Assessment]

This **important** study uses advanced computational methods to elucidate how environmental dielectric properties influence the interaction strengths of tyrosine and phenylalanine in biomolecular condensates. The evidence supporting the claims of the authors is **convincing**, as the simulations are performed rigorously providing mechanistic insights into the origin of the differences between the two aromatic amino acids considered. This study will be of broad interest to researchers studying biomolecular phase separation.

---

## [Referee Report · Reviewer #1 (Public review)]

This is an interesting and timely computational study using molecular dynamics simulation as well as quantum mechanical calculation to address why tyrosine (Y), as part of an intrinsically disordered protein (IDP) sequence, has been observed experimentally to be stronger than phenylalanine (F) as a promoter for biomolecular phase separation. Notably, the authors identified the aqueous nature of the condensate environment and the corresponding dielectric and hydrogen bonding effects as a key to understand the experimentally observed difference. This principle is illustrated by the difference in computed transfer free energy of Y- and F-containing pentapeptides into solvent with various degrees of polarity. The elucidation offered by this work is important. The computation appears to be carefully executed, the results are valuable, and the discussion is generally insightful. However, there is room for improvement in some parts of the presentation in terms of accuracy and clarity, including, e.g., the logic of the narrative should be clarified with additional information (and possibly additional computation), and the current effort should be better placed in the context of prior relevant theoretical and experimental works on cation-π interactions in biomolecules and dielectric properties of biomolecular condensates. Accordingly, this manuscript should be revised to address the following, with added discussion as well as inclusion of references mentioned below.

(1) Page 2, line 61: "Coarse-grained simulation models have failed to account for the greater propensity of arginine to promote phase separation in Ddx4 variants with Arg to Lys mutations (Das et al., 2020)". As it stands, this statement is not accurate, because the cited reference to Das et al. showed that although some coarse-grained model, namely the HPS model of Dignon et al., 2018 PLoS Comput did not capture the Arg to Lys trend, the KH model described in the same Dignon et al. paper was demonstrated by Das et al. (2020) to be capable of mimicking the greater propensity of Arg to promote phase separation than Lys. Accordingly, a possible minimal change that would correct the inaccuracy of this statement in the manuscript would be to add the word "Some" in front of "coarse-grained simulation models ...", i.e., it should read "Some coarse-grained simulation models have failed ...". In fact, a subsequent work [Wessén et al., J Phys Chem B 126: 9222-9245 (2022)] that applied the Mpipi interaction parameters (Joseph et al., 2021, already cited in the manuscript) showed that Mpipi is capable of capturing the rank ordering of phase separation propensity of Ddx4 variants, including a charge scrambled variant as well as both the Arg to Lys and the Phe to Ala variants (see Fig.11a of the above-cited Wessén et al. 2022 reference). The authors may wish to qualify their statements in the introduction to take note of these prior results. For example, they may consider adding a note immediately after the next sentence in the manuscript "However, by replacing the hydrophobicity scales ... (Das et al., 2020)" to refer to these subsequent findings in 2021-2022.

(2) Page 8, lines 285-290 (as well as the preceding discussion under the same subheading & Fig.4): "These findings suggest that ... is not primarily driven by differences in protein-protein interaction patterns ..." The authors' logic in terms of physical explanation is somewhat problematic here. In this regard, "Protein-protein interaction patterns" appears to be a straw man, so to speak. Indeed, who (reference?) has argued that the difference in the capability of Y and F in promoting phase separation should be reflected in the pairwise amino acid interaction pattern in a condensate that contains either only Y (and G, S) and only F (and G, S) but not both Y and F? Also, this paragraph in the manuscript seems to suggest that the authors' observation of similar contact patterns in the GSY and GSF condensates is "counterintuitive" given the difference in Y-Y and F-F potentials of mean force (Joseph et al., 2021); but there is nothing particularly counterintuitive about that. The two sets of observations are not mutually exclusive. For instance, consider two different homopolymers, one with a significantly stronger monomer-monomer attraction than the other. The condensates for the two different homopolymers will have essentially the same contact pattern but very different stabilities (different critical temperatures), and there is nothing surprising about it. In other words, phase separation propensity is not "driven" by contact pattern in general, it's driven by interaction (free) energy. The relevant issue here is total interaction energy or critical point of the phase separation. If it is computationally feasible, the authors should attempt to determine the critical temperatures for the GSY condensate versus the GSF condensate to verify that the GSY condensate has a higher critical temperature than the GSF condensate. That would be the most relevant piece of information for the question at hand.

(3) Page 9, lines 315-316: "...Our ε [relative permittivity] values ... are surprisingly close to that derived from experiment on Ddx4 condensates (45{plus minus}13) (Nott et al., 2015)". For accuracy, it should be noted here that the relative permittivity provided in the supplementary information of Nott et al. was not a direct experimental measurement but based on a fit using Flory-Huggins (FH), but FH is not the most appropriate theory for polymer with long-spatial-range Coulomb interactions. To this reviewer's knowledge, no direct measurement of relative permittivity in biomolecular condensates has been made to date. Explicit-water simulation suggests that relative permittivity of Ddx4 condensate with protein volume fraction ≈ 0.4 can have relative permittivity ≈ 35-50 (Das et al., PNAS 2020, Fig.7A), which happens to agree with the ε = 45{plus minus}13 estimate. This information should be useful to include in the authors' manuscript.

(4) As for the dielectric environment within biomolecular condensates, coarse-grained simulation has suggested that whereas condensates formed by essentially electric neutral polymers (as in the authors' model systems) have relative permittivities intermediate between that of bulk water and that of pure protein (ε = 2-4, or at most 15), condensates formed by highly charge polymers can have relative permittivity higher than that of bulk water [Wessén et al., J Phys Chem B 125:4337-4358 (2021), Fig.14 of this reference]. In view of the role of aromatic residues (mainly Y and F) in the phase separation of IDPs such as A1-LCD and LAF-1 that contain positively and negatively charged residues (Martin et al., 2020; Schuster et al., 2020, already cited in the manuscript), it should be useful to address briefly how the relationship between the relative phase-separation promotion strength of Y vs F and dielectric environment of the condensate may or may not be change with higher relative permittivities.

(5) The authors applied the dipole moment fluctuation formula (Eq.2 in the manuscript) to calculate relative permittivity in their model condensates. Does this formula apply only to an isotropic environment? The authors' model condensates were obtained from a "slab" approach (p.4) and thus the simulation box has a rectangular geometry. Did the authors apply their Eq.2 to the entire simulation box or only to the central part of the box with the condensate (see, e.g., Fig.3C in the manuscript). If the latter is the case, is it necessary to use a different dipole moment formula that distinguishes between the "parallel" and "perpendicular" components of the dipole moment (see, e.g., Eq.16 in the above-cited Wessén et al. 2021 paper). A brief added comments will be useful.

(6) With regard to the general role of Y and F in the phase separation of biomolecules containing positively charged Arg and Lys residues, the relative strength of cation-π interactions (cation-Y vs cation-F) should be addressed (in view of the generality implied by the title of the manuscript), or at least discussed briefly in the authors' manuscript if a detailed study is beyond the scope of their current effort. It has long been known that in the biomolecular context, cation-Y is slightly stronger than cation-F, whereas cation-tryptophan (W) is significantly stronger than either cation-Y and cation-F [Wu & McMahon, JACS 130:12554-12555 (2008)]. Experimental data from a study of EWS (Ewing sarcoma) transactivation domains indicated that Y is a slightly stronger promoter than F for transcription, whereas W is significantly stronger than either Y or F [Song et al., PLoS Comput Biol 9:e1003239 (2013)]. In view of the subsequent general recognition that "transcription factors activate genes through the phase-separation capacity of their activation domain" [Boija et al., Cell 175:1842-1855.e16 (2018)] which is applicable to EWS in particular [Johnson et al., JACS 146:8071-8085 (2024)], the experimental data in Song et al. 2013 (see Fig.3A of this reference) suggests that cation-Y interactions are stronger than cation-F interactions in promoting phase separation, thus generalizing the authors' observations (which focus primarily on Y-Y, Y-F and F-F interactions) to most situations in which cation-Y and cation-F interactions are relevant to biomolecular condensation.

(7) Page 9: The observation of a weaker effective F-F (and a few other nonpolar-nonpolar) interaction in a largely aqueous environment (as in an IDP condensate) than in a nonpolar environment (as in the core of a folded protein) is intimately related to (and expected from) the long-recognized distinction between "bulk" and "pair" as well as size dependence of hydrophobic effects that have been addressed in the context of protein folding [Wood & Thompson, PNAS 87:8921-8927 (1990); Shimizu & Chan, JACS 123:2083-2084 (2001); Proteins 49:560-566 (2002)]. It will be useful to add a brief pointer in the current manuscript to this body of relevant resource in protein science.

Comments on revisions:

The authors have largely addressed my previous concerns and the manuscript has been substantially improved. Nonetheless, it will benefit the readers more if the authors had included more of the relevant references provided in my previous review so as to afford a broader and more accurate context to the authors' effort. This deficiency is particularly pertinent for point number 6 in my previous report about cation-pi interactions. The authors have now added a brief discussion but with no references on the rank ordering of Y, F, and W interactions. I cannot see how providing additional information about a few related works could hurt. Quite the contrary, having the references will help readers establish scientific connections and contribute to conceptual advance.

---

## [Referee Report · Reviewer #2 (Public review)]

Summary:

In this preprint, De Sancho and López use alchemical molecular dynamics simulations and quantum mechanical calculations to elucidate the origin of the observed preference of Tyr over Phe in phase separation. The paper is well written, and the simulations conducted are rigorous and provide good insight into the origin of the differences between the two aromatic amino acids considered.

Strengths:

The study addresses a fundamental discrepancy in the field of phase separation where the predicted ranking of aromatic amino acids observed experimentally is different from their anticipated rankings when considering contact statistics of folded proteins. While the hypothesis that the difference in the microenvironment of the condensed phase and hydrophobic core of folded proteins underlies the different observations, this study provides a quantification of this effect. Further, the demonstration of the crossover between Phe and Tyr as a function of the dielectric is interesting and provides further support for the hypothesis that the differing microenvironments within the condensed phase and the core of folded proteins is the origin of the difference between contact statistics and experimental observations in phase separation literature. The simulations performed in this work systematically investigate several possible explanations and therefore provide depth to the paper.

---

## [Author Response]

The following is the authors’ response to the original reviews.

**Reviewer #1 (Public review):**
This is an interesting and timely computational study using molecular dynamics simulation as well as quantum mechanical calculation to address why tyrosine (Y), as part of an intrinsically disordered protein (IDP) sequence, has been observed experimentally to be stronger than phenylalanine (F) as a promoter for biomolecular phase separation. Notably, the authors identified the aqueous nature of the condensate environment and the corresponding dielectric and hydrogen bonding effects as a key to understanding the experimentally observed difference. This principle is illustrated by the difference in computed transfer free energy of Y- and F-containing pentapeptides into a solvent with various degrees of polarity. The elucidation offered by this work is important. The computation appears to be carefully executed, the results are valuable, and the discussion is generally insightful. However, there is room for improvement in some parts of the presentation in terms of accuracy and clarity, including, e.g., the logic of the narrative should be clarified with additional information (and possibly additional computation), and the current effort should be better placed in the context of prior relevant theoretical and experimental works on cation-π interactions in biomolecules and dielectric properties of biomolecular condensates. Accordingly, this manuscript should be revised to address the following, with added discussion as well as inclusion of references mentioned below.

We are grateful for the referee’s assessment of our work and insightful suggestions, which we address point by point below.

(1) Page 2, line 61: "Coarse-grained simulation models have failed to account for the greater propensity of arginine to promote phase separation in Ddx4 variants with Arg to Lys mutations (Das et al., 2020)". As it stands, this statement is not accurate, because the cited reference to Das et al. showed that although some coarse-grained models, namely the HPS model of Dignon et al., 2018 PLoS Comput did not capture the Arg to Lys trend, the KH model described in the same Dignon et al. paper was demonstrated by Das et al. (2020) to be capable of mimicking the greater propensity of Arg to promote phase separation than Lys. Accordingly, a possible minimal change that would correct the inaccuracy of this statement in the manuscript would be to add the word "Some" in front of "coarse-grained simulation models ...", i.e., it should read "Some coarse-grained simulation models have failed ...". In fact, a subsequent work [Wessén et al., J Phys Chem B 126: 9222-9245 (2022)] that applied the Mpipi interaction parameters (Joseph et al., 2021, already cited in the manuscript) showed that Mpipi is capable of capturing the rank ordering of phase separation propensity of Ddx4 variants, including a charge scrambled variant as well as both the Arg to Lys and the Phe to Ala variants (see Figure 11a of the above-cited Wessén et al. 2022 reference). The authors may wish to qualify their statements in the introduction to take note of these prior results. For example, they may consider adding a note immediately after the next sentence in the manuscript "However, by replacing the hydrophobicity scales ... (Das et al., 2020)" to refer to these subsequent findings in 2021-2022.

We agree with the referee that the wording used in the original version was inaccurate. We did not want to expand too much on the previous results on Lys/Arg, to avoid overwhelming our readers with background information that was not directly relevant to the aromatic residues Phe and Tyr. We have now introduced some of the missing details in the hope that this will provide a more accurate account of what has been achieved with different versions of coarse-grained models. In the revised version, we say the following:

Das and co-workers attempted to explain arginine’s greater propensity to phase separate in Ddx4 variants using coarse-grained simulations with two different energy functions (Das et al., 2020). The model was first parametrized using a hydrophobicity scale, aimed to capture the “stickiness” of different amino acids (Dignon et al., 2018), but this did not recapitulate the correct rank order in the stability of the simulated condensates (Das et al., 2020). By replacing the hydrophobicity scale with interaction energies from amino acid contact matrices —derived from a statistical analysis of the PDB (Dignon et al., 2018; Miyazawa and Jernigan, 1996; Kim and Hummer, 2008)— they recovered the correct trends (Das et al., 2020). A key to the greater propensity for LLPS in the case of Arg may derive from the pseudo-aromaticity of this residue, which results in a greater stabilization relative to the more purely cationic character of Lys (Gobbi and Frenking, 1993; Wang et al., 2018; Hong et al., 2022).

(2) Page 8, lines 285-290 (as well as the preceding discussion under the same subheading & Figure 4): "These findings suggest that ... is not primarily driven by differences in protein-protein interaction patterns ..." The authors' logic in terms of physical explanation is somewhat problematic here. In this regard, "Protein-protein interaction patterns" appear to be a straw man, so to speak. Indeed, who (reference?) has argued that the difference in the capability of Y and F in promoting phase separation should be reflected in the pairwise amino acid interaction pattern in a condensate that contains either only Y (and G, S) and only F (and G, S) but not both Y and F? Also, this paragraph in the manuscript seems to suggest that the authors' observation of similar contact patterns in the GSY and GSF condensates is "counterintuitive" given the difference in Y-Y and F-F potentials of mean force (Joseph et al., 2021); but there is nothing particularly counterintuitive about that. The two sets of observations are not mutually exclusive. For instance, consider two different homopolymers, one with a significantly stronger monomer-monomer attraction than the other. The condensates for the two different homopolymers will have essentially the same contact pattern but very different stabilities (different critical temperatures), and there is nothing surprising about it. In other words, phase separation propensity is not "driven" by contact pattern in general, it's driven by interaction (free) energy. The relevant issue here is total interaction energy or the critical point of the phase separation. If it is computationally feasible, the authors should attempt to determine the critical temperatures for the GSY condensate versus the GSF condensate to verify that the GSY condensate has a higher critical temperature than the GSF condensate. That would be the most relevant piece of information for the question at hand.

We are grateful for this very insightful comment by the referee. We have followed this suggestion to address whether, despite similar interaction patterns in GSY and GSF condensates, their stabilities are different. As in our previous work (De Sancho, 2022), we have run replica exchange MD simulations for both condensates and derived their phase diagrams. Our results, shown in the new Figure 5 and supplementary Figs. S6-S7, clearly indicate that the GSY condensate has a lower saturation density than the GSF condensate. This result is consistent with the trends observed in experiments on mutants of the low-complexity domain of hnRNPA1, where the relative amounts of F and Y determine the saturation concentration (Bremer et al., 2022).

(3) Page 9, lines 315-316: "...Our ε [relative permittivity] values ... are surprisingly close to that derived from experiment on Ddx4 condensates (45{plus minus}13) (Nott et al., 2015)". For accuracy, it should be noted here that the relative permittivity provided in the supplementary information of Nott et al. was not a direct experimental measurement but based on a fit using Flory-Huggins (FH), but FH is not the most appropriate theory for a polymer with long-spatial-range Coulomb interactions. To this reviewer's knowledge, no direct measurement of relative permittivity in biomolecular condensates has been made to date. Explicit-water simulation suggests that the relative permittivity of Ddx4 condensate with protein volume fraction ≈ 0.4 can have a relative permittivity ≈ 35-50 (Das et al., PNAS 2020, Fig.7A), which happens to agree with the ε = 45{plus minus}13 estimate. This information should be useful to include in the authors' manuscript.

We thank the referee for this useful comment. We are aware that the estimate we mentioned is not direct. We have now clarified this point and added the additional estimate from Das et al. In the new version of the manuscript, we say:

Our 𝜀 values for the condensates (39 ± 5 for GSY and 47 ± 3 for GSF) are surprisingly close to that derived from experiments on Ddx condensates using Flory-Huggins theory (45±13) (Nott et al., 2015) and from atomistic simulations of Ddx4 (∼35−50 at a volume fraction of 𝜙 = 0.4) (Das et al., 2020).

(4) As for the dielectric environment within biomolecular condensates, coarse-grained simulation has suggested that whereas condensates formed by essentially electric neutral polymers (as in the authors' model systems) have relative permittivities intermediate between that of bulk water and that of pure protein (ε=2-4, or at most 15), condensates formed by highly charged polymers can have relative permittivity higher than that of bulk water [Wessén et al., J Phys Chem B 125:4337-4358 (2021), Fig.14 of this reference]. In view of the role of aromatic residues (mainly Y and F) in the phase separation of IDPs such as A1-LCD and LAF-1 that contain positively and negatively charged residues (Martin et al., 2020; Schuster et al., 2020, already cited in the manuscript), it should be useful to address briefly how the relationship between the relative phase-separation promotion strength of Y vs F and dielectric environment of the condensate may or may not be change with higher relative permittivities.

We thank the referee for their comment regarding highly charged polymers. However, we have chosen not to address these systems in our manuscript, as they are significantly different from the GSY/GSF peptide condensates under investigation. In polyelectrolyte systems, condensate formation is primarily driven by electrostatic interactions and counterion release, while we highlight the role of transfer free energies. At high dielectric constants (and dielectrics even higher than that of water), the strength of electrostatic interactions will be greatly reduced. In our approach to estimate differences between Y and F, the transfer free energy should plateau at a value of ΔΔG=0 in water. At greater values of ε>80, it becomes difficult to predict whether additional effects might become relevant. As this lies beyond the scope of our current study, we prefer not to speculate further.

(5) The authors applied the dipole moment fluctuation formula (Eq.2 in the manuscript) to calculate relative permittivity in their model condensates. Does this formula apply only to an isotropic environment? The authors' model condensates were obtained from a "slab" approach (page 4) and thus the simulation box has a rectangular geometry. Did the authors apply Equation 2 to the entire simulation box or only to the central part of the box with the condensate (see, e.g., Figure 3C in the manuscript). If the latter is the case, is it necessary to use a different dipole moment formula that distinguishes between the "parallel" and "perpendicular" components of the dipole moment (see, e.g., Equation 16 in the above-cited Wessén et al. 2021 paper). A brief added comment will be useful.

We have calculated the relative permittivity from dense phases only. These dense phases were sliced from the slab geometry and then re-equilibrated. Long simulations were then run to converge the calculation of the dielectric constant. We have clarified this in the Methods section of the paper. We say:

For the calculation of the dielectric constant of condensates, we used the simulations of isolated dense phases mentioned above.

(6) Concerning the general role of Y and F in the phase separation of biomolecules containing positively charged Arg and Lys residues, the relative strength of cation-π interactions (cation-Y vs cation-F) should be addressed (in view of the generality implied by the title of the manuscript), or at least discussed briefly in the authors' manuscript if a detailed study is beyond the scope of their current effort. It has long been known that in the biomolecular context, cation-Y is slightly stronger than cation-F, whereas cation-tryptophan (W) is significantly stronger than either cation-Y and cation-F [Wu & McMahon, JACS 130:12554-12555 (2008)]. Experimental data from a study of EWS (Ewing sarcoma) transactivation domains indicated that Y is a slightly stronger promoter than F for transcription, whereas W is significantly stronger than either Y or F [Song et al., PLoS Comput Biol 9:e1003239 (2013)]. In view of the subsequent general recognition that "transcription factors activate genes through the phase-separation capacity of their activation domain" [Boija et al., Cell 175:1842-1855.e16 (2018)] which is applicable to EWS in particular [Johnson et al., JACS 146:8071-8085 (2024)], the experimental data in Song et al. 2013 (see Figure 3A of this reference) suggests that cation-Y interactions are stronger than cation-F interactions in promoting phase separation, thus generalizing the authors' observations (which focus primarily on Y-Y, Y-F and F-F interactions) to most situations in which cation-Y and cation-F interactions are relevant to biomolecular condensation.

We thank our referee for this insightful comment. While we restrict our analysis to aromatic pairs in this work, the observed crossover will certainly affect other pairs where tyrosine or phenylalanine are involved. We now comment on this point in the discussions section of the revised manuscript. This topic will be explored in detail in a follow-up manuscript we are currently completing. We say:

We note that, although we have not included in our analysis positively charged residues that form cation-π interactions with aromatics, the observed crossover will also be relevant to Arg/Lys contacts with Phe and Tyr. Following the rationale of our findings, within condensates, cation-Tyr interactions are expected to promote phase separation more strongly than cation-Phe pairs.

(7) Page 9: The observation of weaker effective F-F (and a few other nonpolar-nonpolar) interactions in a largely aqueous environment (as in an IDP condensate) than in a nonpolar environment (as in the core of a folded protein) is intimately related to (and expected from) the long-recognized distinction between "bulk" and "pair" as well as size dependence of hydrophobic effects that have been addressed in the context of protein folding [Wood & Thompson, PNAS 87:8921-8927 (1990); Shimizu & Chan, JACS 123:2083-2084 (2001); Proteins 49:560-566 (2002)]. It will be useful to add a brief pointer in the current manuscript to this body of relevant resources in protein science.

We thank the referee for bringing this body of work to our attention. In the revised version of our work, we briefly mention how it relates to our results. We also note that the suggested references have pointed to another of the limitations of our study, that of chain connectivity, addressed in the work by Shimizu and Chan. While we were well aware of these limitations, we had not mentioned them in our manuscript. Concerning the distinction between pair and bulk hydrophobicities, we include the following in the concluding lines of our work:

The observed context dependence has deep roots in the concepts of “pair” and “bulk” hydrophobicity (Wood and Thompson, 1990; Shimizu and Chan, 2002). While pair hydrophobicity is connected to dimerisation equilibria (i.e. the second step in Figure 2B), bulk hydrophobicity is related to transfer processes (the first step). Our work stresses the importance of considering both the pair contribution that dominates at high solvation, and the transfer free energy contribution, which overwhelms the interaction strength at low dielectrics.

**Reviewer #2 (Public review):**
Summary:In this preprint, De Sancho and López use alchemical molecular dynamics simulations and quantum mechanical calculations to elucidate the origin of the observed preference of Tyr over Phe in phase separation. The paper is well written, and the simulations conducted are rigorous and provide good insight into the origin of the differences between the two aromatic amino acids considered.

We thank the referee for his/her positive assessment of our work. Below, we address all the questions raised one by one.

Strengths:The study addresses a fundamental discrepancy in the field of phase separation where the predicted ranking of aromatic amino acids observed experimentally is different from their anticipated rankings when considering contact statistics of folded proteins. While the hypothesis that the difference in the microenvironment of the condensed phase and hydrophobic core of folded proteins underlies the different observations, this study provides a quantification of this effect. Further, the demonstration of the crossover between Phe and Tyr as a function of the dielectric is interesting and provides further support for the hypothesis that the differing microenvironments within the condensed phase and the core of folded proteins is the origin of the difference between contact statistics and experimental observations in phase separation literature. The simulations performed in this work systematically investigate several possible explanations and therefore provide depth to the paper.Weaknesses:While the study is quite comprehensive and the paper well written, there are a few instances that would benefit from additional details. In the methods section, it is unclear as to whether the GGXGG peptides upon which the alchemical transforms are conducted are positioned restrained within the condensed/dilute phase or not. If they are not, how would the position of the peptides within the condensate alter the calculated free energies reported?

The peptides are not restrained in our simulations and can therefore diffuse out of the condensate given sufficient time. Although the GGXGG peptide can, given sufficient time, leave the peptide condensate, we did not observe any escape event in the trajectories we used to generate starting points for switching. Hence, the peptide environment captured in our calculations reflects, on average, the protein-protein and protein-solvent interactions inside the model condensate. We believe this is the right way of performing the calculation of transfer free energy differences into the condensate. We have clarified this point when we describe the equilibrium simulation results in the revised manuscript. We say:

Also, the peptide that experiences the transformation, which is not restrained, must remain buried within the condensate for all the snapshots that we use as initial frames, to avoid averaging the work in the dilute and dense phases.

On the referee’s second point of whether there would be differences if the peptide visited the dilute phase, the answer is that, indeed, we would. We expect that the behaviour of the peptide would approach ΔΔG=0, considering the low protein concentration in the dilute phase. For mixed trajectories with sampling in both dilute and dense phases, our expectation would be a bimodal distribution in the free energy estimates from switching (see e.g. Fig. 8 in DOI:10.1021/acs.jpcb.0c10263). Because we are exclusively interested in the transfer free energies into the condensate, we do not pursue such calculations in this work.

It would also be interesting to see what the variation in the transfer of free energy is across multiple independent replicates of the transform to assess the convergence of the simulations.

Upon submission of our manuscript, we were confident that the results we had obtained would pass the test of statistical significance. We had, after all, done many more simulations than those reported, plus the comparable values of ΔΔG_Transfer_ for both GSY and GSF pointed in the right direction. However, we acknowledge that the more thorough test of running replicates recommended by the referee is important, considering the slow diffusion within the Tyr peptide condensates due to its stickiness. Also, the non-equilibrium switching method had not been tested before for dense phases like the ones considered here.

We have hence followed our referee's suggestion and done three different replicates, 1 μs each, of the equilibrium runs starting from independent slab configurations, for both the GSY and GSF condensates (see the new supporting figures Fig. S1, S2 and S5). We now report the errors from the three replicates as the standard error of the mean (bootstrapping errors remain for the rest of the solvents). Our results are entirely consistent with the values reported originally, confirming the validity of our estimates.

Additionally, since the authors use a slab for the calculation of these free energies, are the transfer free energies from the dilute phase to the interface significantly different from those calculated from the dilute phase to the interior of the condensate?

We thank the referee for this valuable comment, as it has pointed us in the direction of a rapidly increasing body of work on condensate interfaces, for example, as mediators of aggregation, that we may consider for future study with the same methodology. However, as discussed above, we have not considered this possibility in our work, as we decided to focus on the condensate environment, rather than its interface.

The authors mention that the contact statistics of Phe and Tyr do not show significant difference and thereby conclude that the more favorable transfer of Tyr primarily originates from the dielectric of the condensate. However, the calculation of contacts neglects the differences in the strength of interactions involving Phe vs. Tyr. Though the authors consider the calculation of energy contact formation later in the manuscript, the scope of these interactions are quite limited (Phe-Phe, Tyr-Tyr, Tyr-Amide, Phe-Amide) which is not sufficient to make a universal conclusion regarding the underlying driving forces. A more appropriate statement would be that in the context of the minimal peptide investigated the driving force seems to be the difference in dielectric. However, it is worth mentioning that the authors do a good job of mentioning some of these caveats in the discussion section.

We thank the referee for this important comment. Indeed, the similar contact statistics and interaction patterns that we reported originally do not necessarily imply identical interaction energies. In other words, similar statistics and patterns can still result in different stabilities for the Phe and Tyr condensates if the energetics are different. Hence, we cannot conclude that the GSF and GSY condensate environments are equivalent.

To address this point, we have run new simulations for the revised version of our paper, using the temperature-replica exchange method, as before. From the new datasets, we derive the phase diagrams for both the GSF and GSY condensates (see the new Fig. 5). We find that the tyrosine-containing condensate is more stable than that of phenylalanine, as can be inferred from the lower saturation density in the low-density branch of the phase diagram. In consequence, despite the similar contact statistics, the energetics differ, making the saturation density of the GSY slightly lower than that of GSF. This result is consistent with experimental data by Bremer et al (Nat. Chem. 2022).

**Reviewer #3 (Public review):**
Summary:In this study, the authors address the paradox of how tyrosine can act as a stronger sticker for phase separation than phenylalanine, despite phenylalanine being higher on the hydrophobicity scale and exhibiting more prominent pairwise contact statistics in folded protein structures compared to tyrosine.

We are grateful for the referee’s favourable opinion on the paper. Below, we address all of the issues raised.

Strengths:This is a fascinating problem for the protein science community with special relevance for the biophysical condensate community. Using atomistic simulations of simple model peptides and condensates as well as quantum calculations, the authors provide an explanation that relies on the dielectric constant of the medium and the hydration level that either tyrosine or phenylalanine can achieve in highly hydrophobic vs. hydrophilic media. The authors find that as the dielectric constant decreases, phenylalanine becomes a stronger sticker than tyrosine. The conclusions of the paper seem to be solid, it is well-written and it also recognises the limitations of the study. Overall, the paper represents an important contribution to the field.Weaknesses:How can the authors ensure that a condensate of GSY or GSF peptides is a representative environment of a protein condensate? First, the composition in terms of amino acids is highly limited, second the effect of peptide/protein length compared to real protein sequences is also an issue, and third, the water concentration within these condensates is really low as compared to real experimental condensates. Hence, how can we rely on the extracted conclusions from these condensates to be representative for real protein sequences with a much more complex composition and structural behaviour?

We agree with the main weakness identified by the referee. In fact, all these limitations had already been stated in our original submission. Our ternary peptide condensates are just a minimal model system that bears reasonable analogies with condensates, but definitely is not identical to true LCR condensates. The analogies between peptide and protein condensates are, however, worth restating:

(1) The limited composition of the peptide condensates is inspired by LCR sequences (see Fig. 4 in Martin & Mittag, 2018).

(2) The equilibrium phase diagram, showing a UCST, is consistent with that of LCRs from Ddx4 or hnRNPA1.

(3) The dynamical behaviour is intermediate between liquid and solid (De Sancho, 2022).

(4) The contact patterns are comparable to those observed for FUS and LAF1 (Zheng et al, 2020).

The third issue pointed out by the referee requires particular attention. Indeed, the water content in the model condensates is low (~200 mg/mL for GSY) relative to the experiment (e.g. ~600 mg/mL for FUS and LAF-1 from simulations). Considering that both interaction patterns and solvation contribute to the favorability of Tyr relative to Phe, we speculate that a greater degree of solvation in the true protein condensates will further reinforce the trends we observe.

In any case, in the revised version of the manuscript, we have made an effort to insist on the limitations of our results, some of which we plan to address in future work.

**Reviewer #3 (Recommendations for the authors):**
(1) The fact that protein density is so high within GSY or GSF peptide condensates may significantly alter the conclusions of the paper. Can the authors show that for condensates in which the protein density is ~0.2-0.3 g/cm3, the same conclusions hold? Could the authors use a different peptide sequence that establishes a more realistic protein concentration/density inside the condensate?

Unfortunately, recent work with a variety of peptide sequences suggests that finding peptides in the density range proposed by the referee may be very challenging. For example, Pettit and his co-workers have extensively studied the behaviour of GGXGG peptides. In a recent work, using the CHARMM36m force field and TIP3P water, they report densities of ~1.2-1.3 g/mL for capped pentapeptide condensates (Workman et al, Biophys. J. 2024; DOI: 10.1016/j.bpj.2024.05.009). Brown and Potoyan have recently run simulations of zwitterionic GXG tripeptides with the Amber99sb-ILDNQ force field and TIP3P water, starting with a homogenous distribution in cubic simulation boxes (Biophys. J. 2024, DOI: 10.1016/j.bpj.2023.12.027). In a box with an initial concentration of 0.25 g/mL, upon phase separation, the peptide ends up occupying what would seem to be ~1/3 of the box, although we could not find exact numbers. This would imply densities of ~0.75 g/mL in the dense phase, with the additional problem of many charges. Finally, Joseph and her co-workers have recently simulated a set of hexapeptide condensates with varied compositions using a combination of atomistic and coarse-grained simulations. For the atomistic simulations, the Amber03ws force field and TIP4P water were used (see BioRxiv reference 10.1101/2025.03.04.641530). They have found values of the protein density in the dense phase ranging between 0.8 and 1.2 g/mL. The consistency in the range of densities reported in these studies suggests that short peptides, at least up to 7-residues long, tend to form quite dense condensates, akin to those investigated in our work. While the examples mentioned do not comprehensively span the full range of peptide lengths, sequences, and force fields, they nonetheless support the general behaviour we observe. A systematic exploration of all these variables would require an extensive search in parameter space, which we believe falls outside the scope of the present study.

(2) Do the conclusions hold for phase-separating systems that mostly rely on electrostatic interactions to undergo LLPS, like protein-RNA complex coacervates? In other words, could the authors try the same calculations for a binary mixture composed of polyR-polyE, or polyK-polyE?

This is an excellent idea that we may attempt in future work, but the remit of the current work is aromatic amino acids Phe and Tyr only. Hence, we do not include calculations or discussion on polyR-polyE systems in our revised manuscript.

(3) One of the major approximations made by the authors is the length of the peptides within the condensates, which is not realistic, or their density. Specifically, could they double or triple the length of these peptides while maintaining their composition so it can be quantified the impact of sequence length in the transfer of free energies?

We thank the referee for this comment and agree with the main point, which was stated as a limitation in our original submission. The suggested calculations anticipate research that we are planning but will not include in the current work. One of the advantages of our model systems is that the small size of the peptides allows for small simulation boxes and relatively rapid sampling. Longer peptide sequences would require conformational sampling beyond our current capabilities, if done systematically. An example of these limitations is the amount of data that we had to discard from the new simulations we report, which amounts to up to 200 ns of our replica exchange runs in smaller simulation boxes (i.e. >19 μs in total for the 48 replicas of the two condensates!). As stated in the answer to point 1, we have found in the literature work on peptides in the range of 1-7 residues with consistent densities. Additionally, a recent report using alchemical transformations using equilibrium techniques with tetrapeptide condensates, pointing to the role of transfer free energy as driving force for condensate formation, further supports the observations from our work.

Minor issues:

(1) The caption of Figure 3B is not clear. It can only be understood what is depicted there once you read the main text a couple of times. I encourage the authors to clarify the caption.

We have rewritten the caption for greater clarity. Now it reads as follows:

Time evolution of the density profiles calculated across the longest dimension of the simulation box (L) in the coexistence simulations. In blue we show the density of all the peptides, and in dark red that of the F/Y residue in the GGXGG peptide.

(2) Why was the RDF from Figure 5A cut at such a short distance? Can the authors expand the figure to clearly show that it has converged?

In the updated Figure 5 (now Fig. 6), we have extended the g(r) up to r=1.75 nm so that it clearly plateaus at a value of 1.